# Coordinated Regulation of Photosynthesis, Stomatal Traits, and Hormonal Dynamics in *Camellia oleifera* During Drought and Rehydration

**DOI:** 10.3390/biology14080965

**Published:** 2025-08-01

**Authors:** Linqing Cao, Chao Yan, Tieding He, Qiuping Zhong, Yaqi Yuan, Lixian Cao

**Affiliations:** 1Experimental Center of Subtropical Forestry, Chinese Academy of Forestry, Xinyu 338000, China; caolq1991@126.com (L.C.);; 2Key Laboratory of Cultivation and Utilization for Oil-Camellia Resources, Xinyu 338000, China

**Keywords:** *Camellia oleifera*, drought–rehydration cycle, photosynthetic traits, stomatal traits, endogenous hormones, synergistic regulatory network

## Abstract

*Camellia oleifera* is an important woody oil plant in southern China. However, its cultivation in subtropical mountainous regions is often constrained by seasonal droughts. Therefore, it is crucial to evaluate the physiological and ecological response mechanisms of *Camellia oleifera* under drought and rehydration conditions. In this study, we systematically characterized the photosynthetic performance, stomatal morphology, and endogenous hormone responses of two *Camellia oleifera* cultivars with contrasting drought resistance under simulated natural drought–rehydration cycles. The drought-tolerant cultivar exhibited enhanced physiological recovery and a pronounced photosynthetic compensation effect following rehydration, highlighting its superior drought adaptability and physiological resilience. This aids in revealing the mechanism of plant adaptation to drought–rehydration.

## 1. Introduction

Drought is a major abiotic stressor that limits plant growth, development, and geographical distribution [1]. Plants adopt multifaceted resistance to counter water deficit, including drought escape, avoidance, and tolerance mechanisms [2]. Osmotic adjustment under drought stress is achieved by accumulating compatible solutes (e.g., proline, soluble proteins), which help maintain cell turgor. Simultaneously, activation of antioxidant defense systems, including superoxide dismutase (SOD) and peroxidase (POD), mitigates oxidative damage and maintains membrane integrity [3,4,5]. Abscisic acid (ABA) is central in mediating stomatal closure and inducing stress-responsive gene expression among endogenous hormones. Other phytohormones, including auxins (IAA) and gibberellins (GA), also contribute to drought adaptation [6]. As key regulators of CO_2_ uptake and water vapor loss, stomata undergo dynamic adjustments to balance photosynthesis and transpiration under drought conditions [7]. Drought-induced stomatal closure, governed by changes in guard cell wall mechanics, ion channel activity, and cytoskeletal organization [8], reduces water loss but concurrently restricts photosynthetic carbon assimilation, inhibiting growth. Photosynthesis is highly sensitive to drought, with inhibition arising from stomatal and non-stomatal limitations [9]. Under mild stress, reduced stomatal conductance (G_s_) limits CO_2_ availability, lowering the net photosynthetic rate (P_n_). Under moderate-to-severe stress, stomatal and non-stomatal constraints disrupt photochemical activity, often through impaired mesophyll function, Rubisco inactivation, and damage to photosystem II (PSII) [10,11]. Following rehydration, photosynthesis limited by stomatal factors typically recovers quickly, sometimes with compensatory overcorrection. In contrast, the recovery of non-stomatal limitation requires cells to complete damage clearance, molecular regeneration, membrane system reconstruction, and metabolic network rebalancing. These processes are far more complex and time-consuming than regulating stomatal conductance, and the extent of recovery varies depending on the drought resistance of the species [12,13].

*Camellia oleifera*, a major woody oilseed crop in southern China, produces seed oil rich in unsaturated fatty acids and vitamin E, offering significant nutritional, medicinal, and cosmetic value [14]. However, its cultivation in subtropical mountainous regions is frequently constrained by seasonal droughts that impair photosynthetic efficiency, disrupt flower bud differentiation, and compromise fruit development [11,15,16,17], ultimately threatening yield stability and quality. Severe and prolonged drought can even lead to tree mortality. Although previous studies have examined drought effects on *C. oleifera* morphology [18,19], physiology [20,21,22,23], and molecular responses [24,25,26], the following critical knowledge gaps remain. (1) Most investigations address isolated stress phases, overlooking integrated responses among photosynthesis, stomatal behavior, and hormonal regulation during dynamic drought–rehydration cycles. (2) Drought resistance is often assessed using static indicators, lacking systematic analysis of post-rehydration physiological recovery, and (3) the mechanistic basis underlying cultivar-specific differences in drought resistance remains poorly characterized, limiting progress in elite cultivar selection and cultivation strategies.

To address these gaps, we employed two *C. oleifera* cultivars with contrasting drought resistance, ‘CL53’ (tolerant) and ‘CL40’ (sensitive), as model systems. By simulating natural drought–rehydration cycles and integrating analyses of photosynthetic traits, stomatal responses, and endogenous hormone dynamics, we aimed to achieve the following: (1) elucidate the coordinated regulatory network linking photosynthesis, stomatal conductance, and hormone signaling under drought and rehydration; and (2) identify physiological recovery strategies distinguishing drought-tolerant and drought-sensitive cultivars. These findings offer a theoretical framework for evaluating drought resistance, improving water management, and guiding post-drought recovery in *C. oleifera* cultivation.

## 2. Materials and Methods

### 2.1. Study Area and Plant Material

The experiment was conducted at the Experimental Center of Subtropical Forestry, Chinese Academy of Forestry (27°33′–28°08′ N, 114°29′–114°51′ E), located at No. 460, Qianshan West Road, Fenyi County, Xinyu City, Jiangxi Province, China. This region experiences a subtropical humid monsoon climate, characterized by an average annual sunshine duration of 1655.4 h, a mean annual temperature of 17.2 °C (maximum recorded: 40.1 °C), and average annual precipitation of 1594.8 mm. Rainfall distribution is uneven across seasons, with over 60% occurring between April and June, often resulting in recurrent autumn droughts.

Two *C. oleifera* cultivars with contrasting drought resistance were selected, ‘CL53’, a drought-tolerant cultivar with a dwarf growth habit, open canopy architecture, and thick, leathery leaves, and ‘CL40’, a drought-sensitive cultivar characterized by vigorous growth, upright canopy structure, and moderate drought tolerance. Three-year-old container-grown saplings, established in early 2018 under uniform fertilization and irrigation conditions, were used for the study. Twelve healthy and morphometrically uniform individuals per cultivar were selected for experimentation (Table 1).

### 2.2. Experimental Design

The experiment was initiated in July, 2022 to simulate natural drought conditions using mobile rainout shelters that were deployed during rainfall and retracted during sunny periods. Test trees were bordered by protective rows of *C. oleifera* (Figure 1). Soil relative water content (RWC) was monitored weekly at depths of 0–20 cm, 20–40 cm, and 40–60 cm using the ring knife method. After approximately three months, soil RWC stabilized at ~25%, at which point two treatment groups were established—one maintained under continued drought and the other subjected to rehydration. The rehydrated group received an initial irrigation of 100 L per plant, followed by four weekly irrigations of 50 L each, administered between 16:00–17:00. Four treatment groups were thus implemented: drought-stressed CL53 (DS-53), rehydrated CL53 (RW-53), drought-stressed CL40 (DS-40), and rehydrated CL40 (RW-40). As shown in Figure 2, RWC in rehydrated soils was maintained between 50% and 65% for seven days post-irrigation, whereas RWC in drought-stressed soils declined from 25% to ~16%. Photosynthetic and related physiological parameters were measured at 0, 7, 14, 21, and 28 days after rehydration initiation.

### 2.3. Determination of Leaf Relative Water Content

Leaves were sampled from the upper 3–5 whorls of *C. oleifera* spring shoots under similar growth conditions, with three biological replicates and 10 leaves per treatment. Following surface cleaning, fresh weight (W_f_) was recorded immediately. Leaves were then immersed in distilled water for 24 h, blotted dry with absorbent paper, and weighed to obtain the saturated weight (W_t_). Subsequently, leaves were blanched at 105 °C for 15 min, dried at 80 °C to constant weight, and the dry weight (W_d_) recorded. Leaf relative water content (LRWC) was calculated as LRWC (%) = [(W_f_ − W_d_)/(W_t_ − W_d_)] × 100.

### 2.4. Measurement of Photosynthetic Parameters

Photosynthetic parameters were assessed in three *C. oleifera* plants per treatment. Six fully expanded, mature leaves were selected from the upper-middle canopy of each plant. Measurements were conducted between 09:00–11:00 on cloudless days using a Li-6800 portable photosynthesis system (LI-COR, Lincoln, NE, USA), under standardized chamber conditions: photosynthetic photon flux density at 1000 μmol·m^−2^·s^−1^, CO_2_ concentration (C_a_) at 400 μmol·mol^−1^, leaf chamber temperature at 25 °C, and relative humidity maintained between 55% and 65%. Measured parameters included P_n_, transpiration rate (T_r_), G_s_, and intercellular CO_2_ concentration (C_i_). Derived indices included water use efficiency (WUE = P_n_/T_r_) and stomatal limitation (L_s_ = 1 − C_i_/C_a_).

### 2.5. Stomatal Characteristic Measurements

Leaves of comparable developmental stage were harvested from the upper 3–5 whorls of spring shoots on day 28 post-treatment initiation. Lamina segments (5 × 5 mm) adjacent to the midrib (Figure 3) were excised and immediately fixed in FAA solution (70% ethanol:formaldehyde:glacial acetic acid, 90:5:5 *v*/*v*), then stored at 4 °C. After ≥24-h fixation, samples were rinsed three times (15 min each) with 0.1 M phosphate buffer (PB, pH 7.4), post-fixed in 1% osmium tetroxide (in PB) under dark conditions for 1–2 h at room temperature, and rinsed again with PB. Samples were dehydrated in a graded ethanol series (30% to 100%, 15 min per step), transitioned through isopentyl acetate, and subjected to critical point drying. Specimens were gold-coated for 30 s using an ion sputter coater and observed under an SU8100 field-emission scanning electron microscope (Hitachi, Tokyo, Japan). For each treatment, five replicates were examined across three randomly selected fields of view. Stomatal density and aperture dimensions were quantified using Imageview 1.0 software (iRay, Shanghai, China). Stomatal area (S_A_) was calculated assuming elliptical geometry: S_A_ = ^1^/_2_ S_L_ × ^1^/_2_ S_W_ × π, where S_L_ and S_W_ denote stomatal length and width, respectively [27]. Total stomatal pore area per unit leaf area was derived from the product of mean individual stomatal area and open stomatal density.

### 2.6. Endogenous Hormone Assays

Leaves with comparable developmental conditions were sampled from the upper 3–5 whorls of spring shoots. Twenty leaves per treatment were immediately wrapped in aluminum foil, flash-frozen in liquid nitrogen, and stored at −80 °C for subsequent analysis. ABA and gibberellic acid (GA_3_) levels were quantified using high-performance liquid chromatography coupled with tandem mass spectrometry (HPLC-MS/MS) (HPLC: Agilent Technologies Inc., Santa Clara, CA, USA; MS/MS: SCIEX, Framingham, MA, USA) [28], with three biological replicates per treatment.

### 2.7. Data Analysis

Data were analyzed using Microsoft Excel 2016 (Microsoft Corporation, Redmond, WA, USA) and SPSS 26.0 (IBM SPSS, Chicago, IL, USA). One-way ANOVA followed by Duncan’s multiple range test were used to assess differences in photosynthetic parameters, stomatal characteristics, and hormone concentrations among treatments (*p* < 0.05). An independent samples t-test was used to analyze the significant differences in stomatal structure among different varieties (*p* < 0.05). Five function models, exponential, linear, logarithmic, polynomial, and power, were evaluated for fitting ABA and photosynthetic parameter dynamics. Graphical representations were generated using Origin 2021 (OriginLab, Northampton, MA, USA).

## 3. Results

### 3.1. Effects of Drought and Rehydration on Leaf Relative Water Content in C. oleifera

Under drought stress, both *C. oleifera* cultivars exhibited a progressive decline in leaf relative water content (LRWC) (Table 2). No significant reduction in LRWC was observed during the initial 14 days of drought. Compared to initial values (Day 0), DS-53 and DS-40 showed decreases of 6.75% and 3.96%, respectively, on the 28th day of drought. Rehydration triggered significant recovery in both cultivars. CL53 demonstrated rapid restoration, with LRWC increasing from 75.37% to 81.69% within 7 days and subsequently stabilizing. In contrast, CL40 exhibited a more gradual recovery, with LRWC increasing from 75.21% to 83.39%. Throughout drought and rehydration phases (except Day 28), CL53 consistently maintained higher LRWC than CL40, indicating a stronger leaf water retention capacity.

### 3.2. Effects of Drought and Rehydration on Photosynthetic Parameters of C. oleifera

Drought and rehydration significantly influenced photosynthetic performance in both *C. oleifera* cultivars (Figure 4A). During the early drought phase (0–7 days), DS-53 showed a non-significant decline in P_n_ from 4.59 to 4.29 μmol·m^−2^·s^−1^ (−6.5%). Prolonged drought (28 days) led to a further reduction to 3.37 μmol·m^−2^·s^−1^ (–26.6%; *p* < 0.05), indicating cumulative inhibition. DS-40 was more sensitive, with significant reductions at all time points and an overall decline of 32.6% (from 4.42 to 2.98 μmol·m^−2^·s^−1^). CL53 exhibited superior drought tolerance, with attenuated P_n_ reductions and non-significant variation during later drought stages (14–28 days). Upon rehydration, both cultivars showed differential recovery. In RW-40, P_n_ plateaued after day 14 (8.15 μmol·m^−2^·s^−1^), whereas RW-53 exhibited a sustained, phased recovery, reaching 8.99 μmol·m^−2^·s^−1^ by day 21. Thus, CL53 demonstrated dual adaptive advantages: reduced P_n_ inhibition under drought and enhanced post-rehydration recovery.

T_r_ in both cultivars decreased progressively under drought (Figure 4B), stabilizing near 0.5 mmol·m^−2^·s^−1^. Rehydration significantly restored T_r_, with peak recovery observed between days 14 and 21 after rewatering. Compared to initial values (Day 0), RW-53 and RW-40 showed increases of 458.6% and 383.2% at the peak recovery stage (Day 21), respectively.

G_s_ remained strongly suppressed under drought but increased markedly upon rehydration (Figure 4C). Compared to initial values (Day 0), RW-53 and RW-40 showed increases of 348.87% and 313.05%, respectively, on the 28th day of rehydration. G_s_ dynamics closely mirrored T_r_ trends, underscoring the role of stomatal limitation in constraining both gas exchange and transpiration during drought.

C_i_ declined in both cultivars during drought (Figure 4D). In CL53, changes were non-significant, whereas CL40 exhibited a significant 17.95% decrease (from 259.84 to 213.21 μmol·mol^−1^). Rehydration induced a rapid C_i_ increase in CL53 (+13.29% over 0–14 days), followed by a gradual rise, whereas CL40 displayed a slower, steady increase.

L_s_ increased gradually under drought, with significant early elevation (0–14 days) in CL40 but not in CL53 (Figure 4E). Rehydration progressively reduced L_s_, with CL53 showing a larger overall reduction (29.69% vs. 18.55%) compared to initial values (Day 0). CL53 exhibited a more rapid initial decline, significant within 0–14 days, compared to the non-significant, slower decrease observed in CL40. 

WUE declined under drought and, to a lesser extent, during rehydration (Figure 4F). Despite the stress-induced decrease, CL40 consistently maintained higher WUE than CL53 throughout the experiment.

### 3.3. Effects of Drought and Rehydration on Stomatal Characteristics of C. oleifera

Stomata in *C. oleifera* leaves, confined to the abaxial epidermis, regulate transpiration and gas exchange via elliptical to nearly circular apertures controlled by paired guard cells. Significant inter-cultivar differences in stomatal morphology and density were observed (Table 3). CL53 exhibited a lower stomatal density (230.37 mm^−2^, compared to 285.37 mm^−2^ of CL40) but significantly larger stomatal dimensions (length: 24.25 μm, width: 20.19 μm, area: 383.97 μm^2^) than CL40 (21.87 μm, 18.60 μm, 319.91 μm^2^, respectively). Drought–rehydration cycles markedly affected stomatal behavior. Drought stress induced widespread stomatal closure and aperture narrowing (Figure 5), while rehydration effectively restored stomatal functionality.

Prolonged drought significantly reduced both the stomatal density and aperture opening rate in both cultivars (Figure 6A). Under drought, CL53 had an aperture opening rate of 39.67% and a stomatal density of 97.01 mm^−2^, while CL40 exhibited a lower opening rate (33.42%) and slightly reduced density (95.47 mm^−2^); differences between cultivars were not statistically significant. After 21 days of rehydration, stomatal reopening exceeded 98% in both cultivars. However, stomatal density in rehydrated CL40 (280.18 mm^−2^) was significantly higher than in CL53 (219.77 mm^−2^).

Measurements of unit stomatal area under drought and rehydration (Figure 6B) revealed significant treatment-dependent variation. Drought stress substantially reduced stomatal pore area per unit leaf area: in CL53, the area declined to 1900.58 μm^2^·mm^−2^ (a 77.61% reduction vs. RW-53, 8489.94 μm^2^·mm^−2^); in CL40, the value decreased to 2464.15 μm^2^·mm^−2^ (a 76.74% reduction vs. RW-40, 10,594.55 μm^2^·mm^−2^). These results indicate that drought promotes stomatal closure as a water conservation mechanism. Although rehydration significantly restored stomatal area in both cultivars, the extent of recovery was cultivar dependent.

### 3.4. Effects of Drought and Rewatering on Endogenous Hormones in C. oleifera

Under drought conditions, ABA levels significantly increased in both *C. oleifera* cultivars (Table 4). After 28 days of sustained drought, ABA concentrations rose by 22.74% in ‘CL53’ and 30.65% in ‘CL40’. Following rewatering, ABA levels progressively declined, stabilizing after 21 days with reductions of 63.05% (CL53) and 66.78% (CL40) relative to drought-induced peaks. This dynamic modulation underscores the central role of ABA in coordinating drought stress responses and recovery.

Concurrently, GA_3_ content declined gradually in both cultivars under drought stress (Table 5), although changes were not statistically significant until Day 21. After rewatering, GA_3_ levels increased steadily, stabilizing after 14 days. Notably, CL53 exhibited a more rapid recovery in GA_3_ content, reflecting superior drought resilience.

### 3.5. Correlation Between Photosynthetic Characteristics and ABA Content in C. oleifera

Given the pronounced responsiveness of ABA to drought and rewatering, and its established role in regulating stomatal aperture through signal transduction pathways, we further evaluated the relationship between ABA content and photosynthetic parameters in *C. oleifera* (Figure 7). Five regression models (exponential, linear, logarithmic, polynomial, power) were tested, with the best-fitting model selected for each cultivar-variable combination.

A significant negative correlation was observed between ABA content and P_n_ (Figure 7A). For CL53, the data best fit an exponential decay model (y = −2.8095 + 15.8521 × e^−0.0296x^; R^2^ = 0.9212), while CL40 followed a polynomial model (y = 11.5058 − 0.2756x + 0.0019x^2^; R^2^ = 0.9629). The sharper decline in P_n_ in CL40 suggests greater ABA sensitivity compared to CL53.

T_r_ also exhibited significant negative correlations with ABA in both cultivars, conforming to power function models (Figure 7B): CL53 (y = 267.5508 × x^–1.8375^; R^2^ = 0.7896) and CL40 (y = 77.9072 × x^−1.4251^; R^2^ = 0.9763). The stronger fit in CL40 indicates a more pronounced ABA-mediated reduction in T_r_.

G_s_ was significantly negatively correlated with ABA content (Figure 7C), with a power function best describing the trend in CL53 (y = 6.2072 × x^−1.5224^; R^2^ = 0.8596) and an exponential model fitting CL40 (y = 0.0209 + 0.3334 × 0.9103^x^; R^2^ = 0.9202).

C_i_ declined significantly with increasing ABA (Figure 7D), modelled linearly in CL53 (y = 329.9571 − 3.1528x; R^2^ = 0.9401) and polynomially in CL40 (y = 273.4619 + 1.0372x − 0.0482x^2^; R^2^ = 0.8909).

A significant positive correlation was observed between ABA and L_s_ (Figure 7E), fitting a linear model for CL53 (y = 0.1751 + 0.0079x; R^2^ = 0.9401) and a polynomial model for CL40 (y = 0.3163 − 0.0026x + 1.2054 × 10^−4^x^2^; R^2^ = 0.8909).

The relationship between ABA content and WUE followed polynomial trends in both cultivars (Figure 7F): CL53 (y = −9.8040 + 1.4304x − 0.0288x^2^; R^2^ = 0.8158) and CL40 (y = −2.0233 + 0.6158x − 0.0088x^2^; R^2^ = 0.8722). The parabolic curves suggest that WUE initially increased then declined with rising ABA concentrations, with CL40 attaining peak WUE at higher ABA levels than CL53.

## 4. Discussion

Drought is a predominant abiotic stressor affecting plant development and productivity [29,30]. Although *C. oleifera* is considered drought-tolerant, its cultivation in southern China’s hilly and mountainous regions is frequently challenged by hot, dry summers that limit growth and yield [31,32]. Plants exhibit intrinsic adaptive mechanisms under drought, enabling physiological and biochemical assessment of stress responses [33].

Photosynthesis, the central physiological process supplying carbon and energy substrates, is highly susceptible to drought-induced disruption of carbon metabolism [34,35,36]. In this study, drought markedly inhibited photosynthetic capacity in both *C. oleifera* cultivars, with distinct varietal responses reflecting underlying mechanistic complexity. During early drought (0–7 days), reduced P_n_ was primarily attributable to decreased G_s_, indicating that stomatal limitation predominated [11]. After 28 days of sustained drought, the reduction in P_n_ was lower in CL53 (26.6%) than in CL40 (32.6%). The lack of significant changes in C_i_ indicates that the observed reduction in photosynthesis was predominantly attributable to impaired photosynthetic capacity within mesophyll cells. In contrast, the P_n_ of CL53 showed only a minor decrease, indicating that the mesophyll cells of CL53 could still sustain a high level of photosynthetic capacity under drought conditions [37].

Post-rehydration, recovery of P_n_ diverged between cultivars: CL53 showed a progressive increase, reaching 8.99 μmol·m^−2^·s^−1^, whereas CL40 plateaued after 14 days. The higher recovery rate and evident photosynthetic compensation in CL53 [38] contrast with the delayed response in CL40, which may reflect more severe chloroplast ultrastructural damage [9]. C_i_ dynamics in CL53 further indicated more complete alleviation of stomatal limitation, consistent with its greater drought resistance. Although WUE decreased during drought in both cultivars, CL40 consistently exhibited higher WUE than CL53, possibly because of greater stomatal pore density and more precise transpiration control [7]. However, this advantage did not translate to stronger drought resistance, suggesting that sustained photosynthetic carbon assimilation outweighs short-term water conservation during prolonged drought [39,40,41].

As key gas exchange regulators, stomata play a central role in drought adaptation [7,42]. In *C. oleifera*, drought induced pronounced stomatal closure, an adaptive response that restricts CO_2_ diffusion and limits photosynthesis [43]. Upon rehydration, both cultivars achieved >98% stomatal aperture reopening, but the recovery of open stomatal density and pore area per unit leaf surface differed. CL40 exhibited higher post-rehydration stomatal density and larger unit area pore size, which may facilitate more rapid gas exchange and photosynthetic recovery [44]. Differences in stomatal density may also impact plant responses to drought stress [45]. However, significant inter-varietal differences in stomatal morphology were observed. CL53 had fewer but larger stomata (greater length, width, and area) than CL40. These traits likely underlie cultivar-specific differences in stomatal regulation and drought responses [46], with drought-tolerant cultivars potentially optimizing stomatal dimensions to maintain higher conductance and photosynthetic rates under stress [47].

Endogenous phytohormones play essential roles in drought signaling, modulating synthesis, transport, and hormonal balance to regulate adaptive responses [33,48]. ABA accumulation is a hallmark of drought response, promoting stomatal closure and restricting growth and metabolism [49]. Drought significantly increased ABA levels in both cultivars, consistent with its role as a central regulator of stomatal behavior and stress-responsive gene expression [50,51]. Interestingly, the drought-sensitive CL40 accumulated more ABA than CL53, possibly reflecting excessive signaling that disrupts physiological coordination [52]. Following rehydration, ABA levels declined rapidly in both cultivars, with CL53 exhibiting faster ABA degradation and more pronounced GA_3_ recovery. This suggests that drought-tolerant cultivars more effectively reactivate gibberellin biosynthesis to support cell elongation and photosynthetic recovery [53]. GA_3_, an essential hormone for promoting plant growth, opposes ABA in the plant’s response to drought stress. GA_3_ facilitates stomatal opening, boosts CO_2_ absorption, and enhances photosynthetic efficiency [54]. This study revealed that drought induced a progressive decrease in GA_3_ levels in both varieties of *C. oleifera*, which was followed by a gradual increase upon rehydration. However, the extent of these changes was relatively minor, suggesting a less significant regulatory role for GA_3_ in photosynthetic capacity. Although CL40 maintained higher GA_3_ concentrations, potentially supporting vigorous growth, elevated GA_3_ under drought may exacerbate carbon expenditure and impair drought tolerance, contrasting with ABA-mediated conservation responses.

Significant correlations were identified between ABA and photosynthetic parameters: negative correlations with P_n_, T_r_, G_s_, and C_i_, consistent with previous findings [13], and a parabolic relationship with WUE (initial increase followed by decline). These relationships reinforce the regulatory role of ABA in *C. oleifera* photosynthesis. Varietal differences in ABA–photosynthesis correlations further reflect inherent divergence in drought resistance mechanisms and adaptive physiological strategies.

## 5. Conclusions

This study systematically characterized the photosynthetic performance, stomatal morphology, and endogenous hormone responses of two *C. oleifera* cultivars with contrasting drought resistance under simulated natural drought–rehydration cycles. The drought-tolerant cultivar exhibited enhanced physiological recovery and a marked photosynthetic compensation effect following rehydration, underscoring its superior drought adaptability and physiological resilience. These findings provide a theoretical basis for the selection of drought-tolerant cultivars and the implementation of precision water management practices. Future research should focus on elucidating the signal transduction pathways and gene regulatory networks underlying cultivar-specific drought responses. Additionally, integrating these insights with climate change models and water-use optimization strategies will be essential for establishing comprehensive frameworks to support the sustainable development of the *C. oleifera* industry.

## Figures and Tables

**Figure 1 biology-14-00965-f001:**
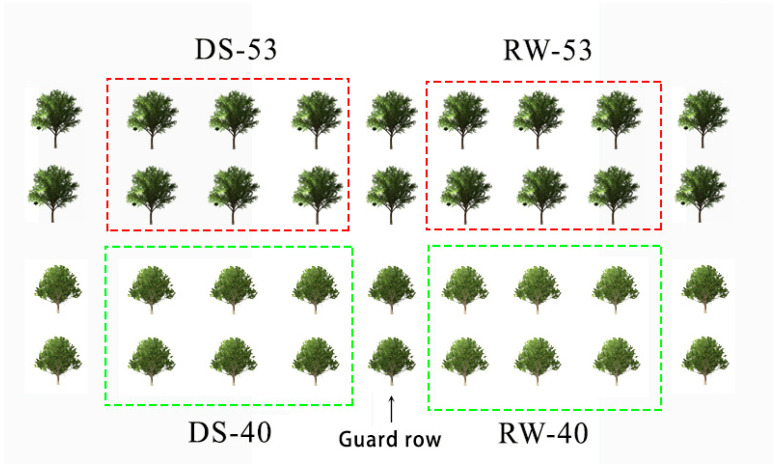
Experimental design of the drought–rehydration treatment.

**Figure 2 biology-14-00965-f002:**
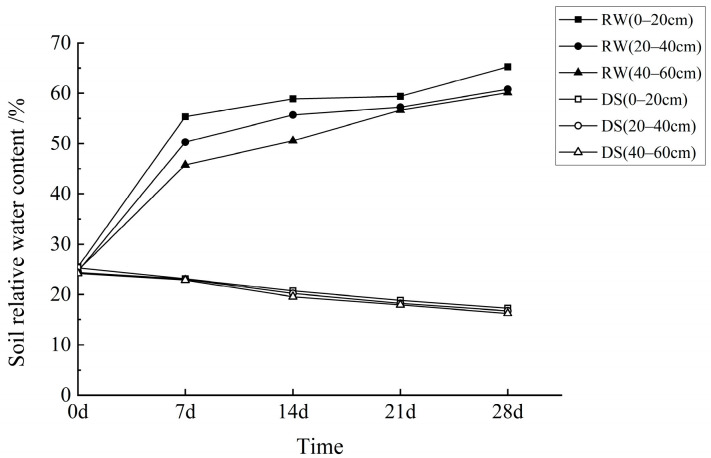
Changes in relative soil water content in *Camellia oleifera* under drought and rehydration treatments. DS, drought-stressed group; RW, rehydrated group. Soil layers represent depths of 0–20 cm, 20–40 cm, and 40–60 cm, respectively.

**Figure 3 biology-14-00965-f003:**
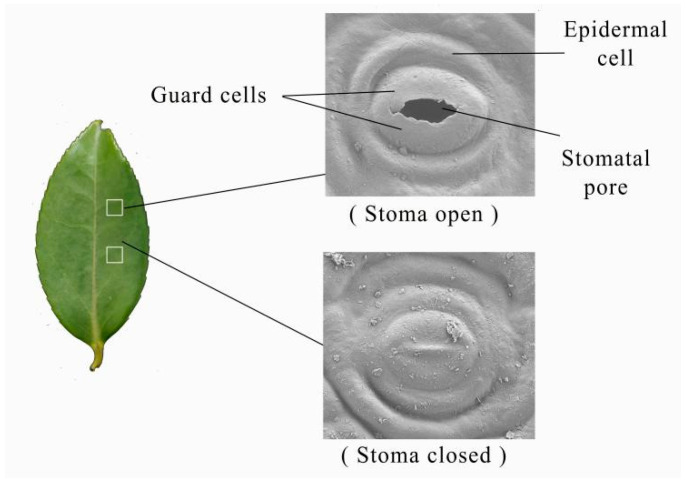
Stomatal morphology of *C. oleifera* leaf surfaces.

**Figure 4 biology-14-00965-f004:**
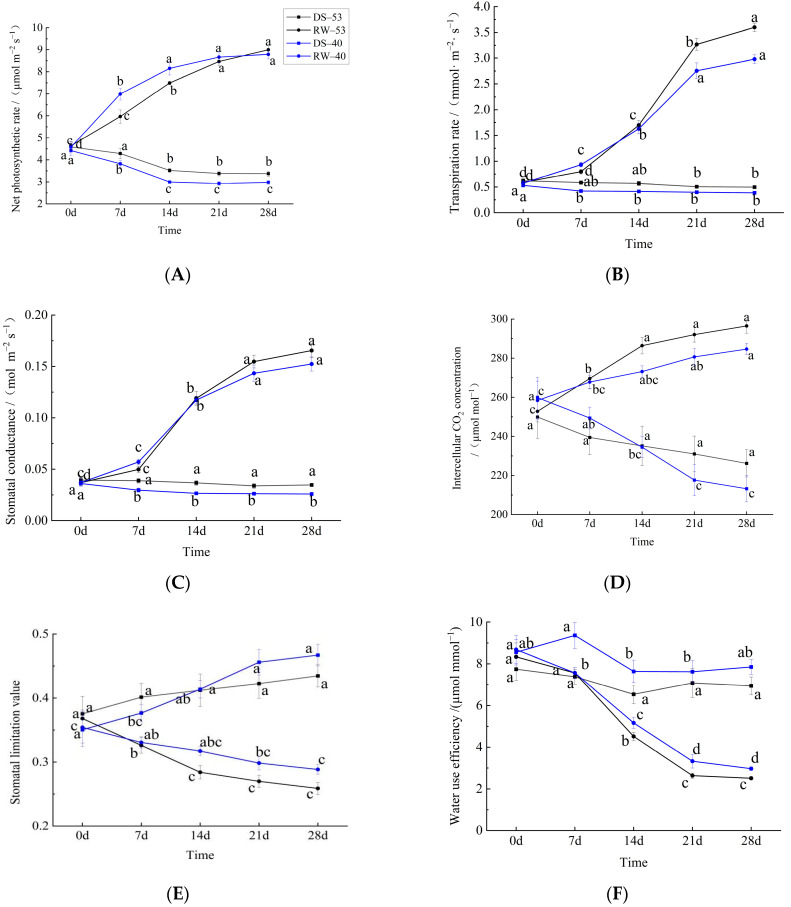
Photosynthetic responses of *C. oleifera* leaves to drought and rewatering. (**A**) Net photosynthetic rate (P_n_). (**B**) Transpiration rate (T_r_). (**C**) Stomatal conductance (G_s_). (**D**) Intercellular CO_2_ concentration (C_i_). (**E**) Stomatal limitation value (L_s_). (**F**) Water use efficiency (WUE). Different lowercase letters indicate significant differences within the same treatment across time points (*p* < 0.05).

**Figure 5 biology-14-00965-f005:**
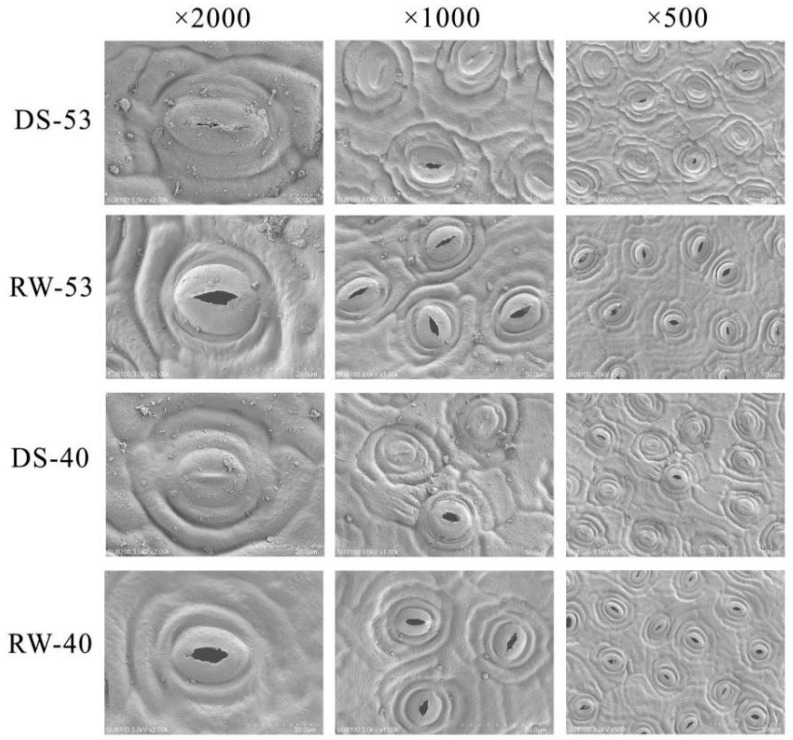
Representative scanning electron microscopy (SEM) images of stomata on *C. oleifera* leaves after 28 days of drought or rewatering treatment. Images show stomatal morphology at ×2000, ×1000, and ×500 magnifications (left to right).

**Figure 6 biology-14-00965-f006:**
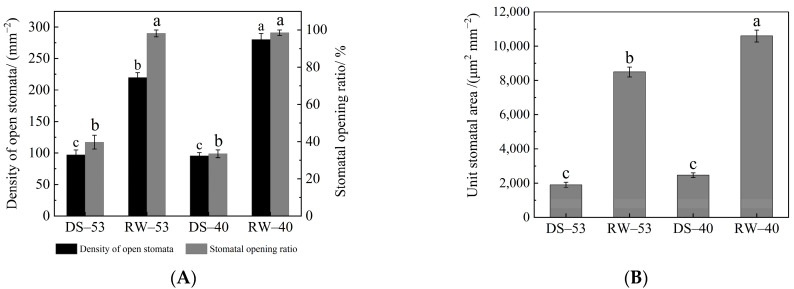
Effects of drought and rewatering on stomatal traits of *C. oleifera* leaves. (**A**) Density of open stomata (left) and stomatal opening ratio (right). (**B**) Unit stomatal area. Different lowercase letters indicate significant differences between treatments (*p* < 0.05).

**Figure 7 biology-14-00965-f007:**
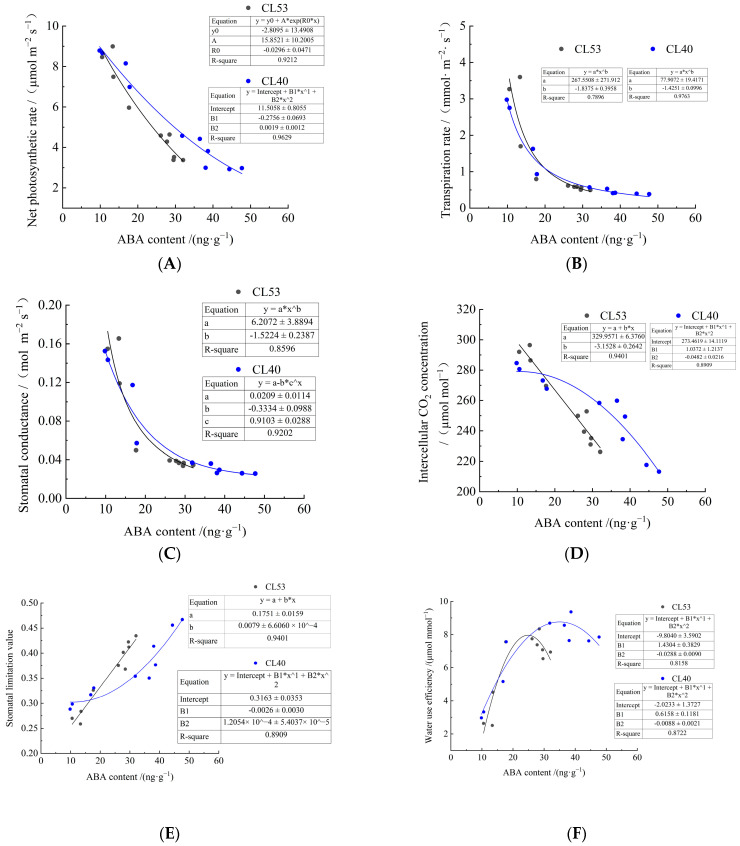
Correlation between abscisic acid (ABA) content and photosynthetic parameters in *C. oleifera* cultivars under drought–rewatering treatment. (**A**) Net photosynthetic rate (P_n_). (**B**) Transpiration rate (T_r_). (**C**) Stomatal conductance (G_s_). (**D**) Intercellular CO_2_ concentration (C_i_). (**E**) Stomatal limitation value (L_s_). (**F**) Water use efficiency (WUE).

**Table 1 biology-14-00965-t001:** Basic information on the *Camellia oleifera* test cultivars.

Variety	Forest Age (a)	Average Base Diameter (cm)	Average Tree Height (cm)	Average Crown Width (cm)
CL53	7	6.2 ± 0.40	135.4 ± 11.3	162
CL40	7	5.7 ± 0.35	172.8 ± 13.5	134

**Table 2 biology-14-00965-t002:** Effects of drought and rehydration on leaf relative water content in *C. oleifera*.

Treatment	Leaf Relative Water Content (%)				
	0 d	7 d	14 d	21 d	28 d
DS-53	76.61 ± 0.64 ab	75.95 ± 1.02 ab	77.44 ± 1.11 a	74.17 ± 1.80 b	71.43 ± 2.26 b
RW-53	75.37 ± 1.28 b	81.69 ± 1.34 a	81.46 ± 0.75 a	81.36 ± 1.17 a	82.48 ± 0.85 a
DS-40	75.54 ± 0.50 a	75.86 ± 0.61 a	74.08 ± 1.24 ab	72.27 ± 2.11 b	72.54 ± 1.29 b
RW-40	75.21 ± 0.66 d	78.34 ± 1.38 c	80.19 ± 1.39 bc	81.24 ± 1.87 b	83.39 ± 1.00 a

In each row, different lowercase letters indicate statistically significant differences at *p* < 0.05.

**Table 3 biology-14-00965-t003:** Stomatal characteristics of different *C. oleifera* cultivars.

Variety	Stomatal Density (mm^−2^)	Stomatal Apparatus Length (μm)	Stomatal Aperture Width (μm)	Stomatal Apparatus Area (μm^2^)
CL53	230.37 ± 11.94 b	24.25 ± 1.39 a	20.19 ± 1.49 a	383.97 ± 32.47 a
CL40	285.37 ± 12.23 a	21.87 ± 1.06 b	18.60 ± 1.88 b	319.91 ± 40.43 b

Different lowercase letters indicate statistically significant differences at *p* < 0.05 in each column.

**Table 4 biology-14-00965-t004:** Effects of abscisic acid levels in *C. oleifera* leaves under drought and rehydration conditions.

Treatment	ABA Content/(ng·g^−1^)				
	0 d	7 d	14 d	21 d	28 d
DS-53	26.12 ± 0.73 d	27.77 ± 0.52 c	29.65 ± 1.21 b	29.50 ± 0.49 b	32.06 ± 1.01 a
RW-53	28.45 ± 0.89 a	17.66 ± 0.46 b	13.51 ± 0.90 c	10.51 ± 0.51 d	13.33 ± 0.55 c
DS-40	36.51 ± 0.84 d	38.65 ± 0.30 c	38.04 ± 1.09 c	44.36 ± 1.02 b	47.70 ± 0.69 a
RW-40	31.83 ± 0.36 a	17.82 ± 0.72 b	16.79 ± 0.73 c	10.57 ± 0.58 d	9.83 ± 0.17 d

In each row, different lowercase letters indicate statistically significant differences at *p* < 0.05.

**Table 5 biology-14-00965-t005:** Effects of drought and rehydration on gibberellin levels in *C. oleifera* leaves.

Treatment	GA_3_ Content/(ng·g^−1^)				
	0 d	7 d	14 d	21 d	28 d
DS-53	2.96 ± 0.12 a	2.79 ± 0.15 a	2.68 ± 0.13 ab	2.66 ± 0.13 ab	2.42 ± 0.28 b
RW-53	2.58 ± 0.11 c	3.39 ± 0.16 b	3.98 ± 0.08 a	3.80 ± 0.19 a	3.89 ± 0.36 a
DS-40	6.77 ± 0.31 a	6.58 ± 0.30 a	6.49 ± 0.24 a	6.50 ± 0.29 a	5.91 ± 0.34 b
RW-40	6.42 ± 0.25 b	6.45 ± 0.24 b	7.15 ± 0.27 a	7.43 ± 0.18 a	7.51 ± 0.18 a

In each row, different lowercase letters indicate statistically significant differences at *p* < 0.05.

## Data Availability

The data presented in this study are available within the article.

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
