# Peer review of "Coordinated Regulation of Photosynthesis, Stomatal Traits, and Hormonal Dynamics in *Camellia oleifera* During Drought and Rehydration"

_biology, 2025, doi:10.3390/biology14080965_

Round 1
Reviewer 1 Report
Comments and Suggestions for Authors
Line187 There is an error in the citation. It should be Table 2, not 4
Line188 The statement is incorrect. Check carefully
Line193 The statement is incorrect. CL40 consistently maintains a higher RWC than CL53
Line212 The statement is not accurate. It depends on which day, the percentage is calculated from day 0 or day 7
Line216 The statement is not accurate. It depends on which day, the percentage is calculated on day 0 or day 7
Line222 The expression is wrong. According to the figure, CL35 should consistently maintain a higher WUE than CL40 throughout the experiment under drought conditions.
Line227 It also does not indicate which day the overall decline is larger and the percentage figure derived from it. This is inaccurate
Line230 The F graph should show water use efficiency, and the figure caption shows intercellular carbon dioxide concentration
Line237 It mentions a significant improvement that the table data does not reflect significance abc and it should be that the table data corresponds to the text
Line334 Suggestion: The writing is messy and it is suggested that there should be some logic
Line342 The statement is wrong all the time. According to the figure, CL40 is lower than CL53 on day 21, so the explanation is not correct. It is suggested to rewrite it.
The author should read the manuscript carefully to ensure that every part is reasonable and understandable to the researcher, including language descriptions, references, abbreviations, etc
Author Response
Thank you very much for taking the time to review this manuscript. Those comments are very helpful for revising and improving our paper, as well as the important guiding significance to our research. We have studied the comments carefully and have made a correction which we hope meet with approval. The following replies are the main comments made to you, and we would like to express our thanks to you once again.
For specific modifications, please refer to the attachment.

Reviewer 2 Report
Comments and Suggestions for Authors
The manuscript by Cao et al. considers the drought resistance of two varieties of camelia oleifera. The investigation seems to be interesting. However, I have some comments.
- The non-stomatal limitations of conductivity for CO2 should be described in more detail in introduction. Why the recovery of non-stomatal conductivity needs the long-time intervals?
- Table 3. The significances should be added.
- Lines 276-278. The sentence should be clarified.
- Why correlation between photosynthetic parameters and GA3 concentration was not investigated?
- The influence of ABA and GA3 on stomatal conductivity and photosynthesis should be described in more detail.
Author Response

(The authors gave the same response as above.)

Round 2
Reviewer 1 Report
Comments and Suggestions for Authors
no